# Genome-Wide Association Study (GWAS) for Mesocotyl Elongation in Rice (*Oryza sativa* L.) under Multiple Culture Conditions

**DOI:** 10.3390/genes11010049

**Published:** 2019-12-31

**Authors:** Hongyan Liu, Junhui Zhan, Jiaolong Li, Xiang Lu, Jindong Liu, Yamei Wang, Quanzhi Zhao, Guoyou Ye

**Affiliations:** 1CAAS-IRRI Joint Laboratory for Genomics-Assisted Germplasm Enhancement, Agricultural Genomics Institute in Shenzhen, Chinese Academy of Agricultural Sciences, Shenzhen 518120, China; hongyanliu@caas.cn (H.L.); jhzhanhau@126.com (J.Z.); lijiaolong@caas.cn (J.L.); luxiang@caas.cn (X.L.); Liujindong@caas.cn (J.L.); g.ye@irri.org (G.Y.); 2Collaborative Innovation Center of Henan Grain Crops and Key Laboratory of Rice Biology in Henan Province, College of Agronomy, Henan Agricultural University, Zhengzhou 450002, China; 3Strategic Innovation Platform, International Rice Research Institute, DAPO Box 7777, Metro Manila, Philippines

**Keywords:** dry direct-seeded rice, early vigor, QTL, candidate gene, phenotyping

## Abstract

Mesocotyl is a crucial organ for pushing buds out of soil, which plays a vital role in seedling emergence and establishment in dry direct-seeded rice. However, the genetic mechanisms of mesocotyl elongation remains unclear. In our study, 208 rice accessions were used to identify the SNPs significantly associated with mesocotyl length under various culture conditions, including sand, water and soil. The mesocotyl length ranges from 0 to 4.88 cm, 0 to 3.99 cm and 0 to 4.51 cm in sand, water and soil covering, respectively. A total of 2,338,336 SNPs were discovered by re-sequencing of 208 rice accessions. Genome-wide association study (GWAS) based on mixed linear model (MLM) was conducted and 16 unique loci were identified on chromosomes 1, 2 (2), 3, 4, 5 (2), 6 (2), 7, 8, 9 (2) and 12 (3), respectively, explaining phenotypic variations ranging from 6.3 to 15.9%. Among these loci, 12 were stable across two or more environments. Ten out of the sixteen loci coincided with known genes or quantitative trait locus (QTL), whereas the other six were potentially novel loci. Furthermore, five high-confidence candidate genes related to mesocotyl elongation were identified on chromosomes 1, 3, 5, 9 and 12. Moreover, qRT-PCR analysis showed that all the five genes showed significant expression difference between short-mesocotyl accessions and long-mesocotyl accessions. This study provides new insights into the genetic architecture of rice mesocotyl, the associated SNPs and germplasms with long mesocotyl could be useful in the breeding of mechanized dry direct-seeded rice.

## 1. Introduction

Rice (*Oryza sativa*) is one of the most important cereal crops grown worldwide. Dry direct-seeded rice refers to the process of establishing the crop from seeds sown on non-puddled and unsaturated soil; in contrast, seedlings from nursery are transplanted into puddled or submerged soil in transplanted rice [1]. Compared with traditional transplanted rice, dry direct-seeded rice has been proposed as a water-efficient and labor-saving approach, which can reduce the cost of water and labor at about 50% [2,3]. Also, dry direct-seeded rice could efficiently utilize early-season rainfall in drought-prone environments and complete its growth cycle within the wet season in rainfed lowlands [4]. To date, direct seeding has been adopted by more than 25% of the worldwide rice cultivation area [1]. Dry direct-seeded rice is becoming a popular option in Philippines, India, Thailand, Cambodia, Laos and Indonesia in tropical Asia as well as the US, Australia and Latin America [5,6].

However, a few constraints in dry direct-seeded rice, e.g., poor seedling establishment, weed infestation and lodging susceptibility could lead to large reduction in grain yield and quality [7,8,9,10,11]. Seedling establishment is very important for high yielding and controlling weed infestation in direct seeding rice [10]. Meanwhile, deep sowing could be good for root and basal internodes elongation in deeper soil, which could gain the stability of rice plant and increase lodging resistance of direct seeding rice [12]. However, deep sowing is always in contradiction to seedling uniformity and leads to grain yield reduction [13]. Rapid and good seedling establishment is important for weed competitiveness and good harvesting. Mesocotyl, an organ between the coleoptilar node and the basal part of seminal root in young monocot seedlings, plays a key role in pushing buds out of the deep water or soil during germination for successful seedling establishment. Longer mesocotyl can facilitate seedling establishment under deep sown conditions in dry direct-seeded rice [14,15,16,17,18]. Besides, Mgonja et al. (1988) also reported the strong association between mesocotyl length and seedling vigor [19]. However, the molecular mechanisms of rice mesocotyl length variation are poorly understood.

Genome-wide association study (GWAS) based on linkage disequilibrium (LD) has been widely adopted to identify loci significantly associated with important and complex agronomic traits in rice [20,21,22]. Extremely high resolution can be achieved by dense SNPs identified in diverse germplasm panels based on the next-generation sequencing (NGS) or SNP chip approaches [20,21,23,24]. Thousands of rice landraces or cultivars have been screened for mesocotyl length by GWAS and dozens of QTL have been reported in previous studies [17,18,25,26]. Wu et al. (2015) screened 270 rice accessions and 16 loci were identified associated with mesocotyl elongation [17]. Furthermore, 469 Indica accessions were used to measure mesocotyl length, and 23 loci were significantly associated with mesocotyl length [18]. Zhao et al. (2018) evaluated the mesocotyl length of 621 rice accessions and detected 13 QTLs [26]. Also, Sun et al. (2018) have identified three QTL for mesocotyl length from 510 rice accessions [25].

Identifying the QTLs in multi-environments could provide accurate information for gene cloning and molecular breeding. In this study, GWAS based on resequencing was conducted in a set of 208 rice accessions for mesocotyl elongation under sand, water and soil (2.0 cm, 4.0 cm and 6.0 cm). The objectives of this study were: (1) To dissect the genetic architecture of mesocotyl elongation, (2) to identify SNPs significantly associated with mesocotyl length, (3) to search for candidate mesocotyl elongation genes for further study, and (4) to select accessions with longer mesocotyl for the breeding of mechanized dry direct-seeded rice.

## 2. Materials and Methods

### 2.1. Plant Materials

The panel for GWAS consisted of two parts of rice germplasms, one part (114 accessions) includes advanced lines from International Rice Research Institute (IRRI) and mega varieties released in southeast Asian countries, such as India, Philippines and Bangladesh. The others (94 accessions) are landraces and released varieties mainly from China. All the 208 accessions could be divided into two ecotypes, *Indica* (170) and *Japonica* (38) (Appendix A).

### 2.2. Genotyping

Total genomic DNA for re-sequencing was extracted from young leaves according to the CTAB method. The 208 accessions were genotyped using the Illumina HiSeq 2000 (PE150) (50X) by Berry Genomics Corporation (https://www.berrygenomics.com/) (Beijing, China). The average sequencing depth of each accession genome was 50×. Reads were aligned to the Nipponbare RefSeq (IRGSP-1.0) using BWA-MEM (release 0.7.10). Then, the mapped reads were sorted and duplicated were removed by Picard tools (http://broadinstitute.github.io/picard/). The variants for each accession were called by the GATK best practices (release 3.2-2).

### 2.3. Measurement of Mesocotyl Length

To evaluate the variation of mesocotyl length (the distance from the basal part of seminal root to the coleoptilar node) at varied sowing depths, two replications were set up, and in each replication 15 good-quality seeds from each accession were sown at a depth of 2, 4 and 6 cm in soil contained by plastic trays. The size of plastic tray contained 50 holes with the size of 9.5 cm depth, 4.5 cm top diameter and 2.1 cm bottom diameter. After sowing, the plastic tray was kept in a plastic pallet with 3-cm-deep soil, and the whole system were maintained in a 30 °C dark incubator and the soil in each pallet was kept water saturated for seed germination and seedling growth. Emergence rate was recorded every day, until the emergence rate of any accession reached 100%. Three days after that, seedlings from each hole were carefully excavated and washed for measuring mesocotyl length using Image J (https://imagej.en.softonic.com/).

To quantify the mesocotyl length under different culture media, sand and water culture were used, in addition to soil culture. Fifteen good-quality seeds were sown at a depth of 6 cm in sand, the culture method and sampling timing in sand was the same as that in soil. As for water culture, 15 good-quality seeds were put on a gauze in each hole of a plastic tray (50 holes with a depth of 4.8 cm, top diameter at 4.1 cm and bottom diameter at 2.1 cm). The plastic tray was put on a plastic pallet containing water in level of 2 cm from the bottom. Each plastic tray was wrapped with tin foil paper to avoid illumination and kept in phytotron with constant temperature of 30 °C for 10 days. After that, ten uniform seedlings out of 15 seeds of each accession were used for mesocotyl length measurement with Image J. Two replications were set up in both culture methods and mean of the two replications was used for GWAS analysis.

### 2.4. Population Structure and LD Decay Analysis

The population structure of the 208 accessions was analyzed using Admixture 1.3.0 [27] based on 2000 SNP markers with default cross-validation (K value ranged from 2 to 5). An adhoc quantity statistic ΔK was used to predict the real number of subpopulations and k = 2 was chosen. Besides, the principal component analysis (PCA) and neighbor-jointing trees were also used to validate population stratification with the software Tassel v5.0 (https://www.maizegenetics.net/tassel).

LD decay analysis was done for the whole population. LD decay was measured by correlation coefficients (*r*^2^) for all pairs of SNPs within 500 Kb that were calculated using PopLDdecay v3.27 (https://github.com/BGI-shenzhen/PopLDdecay) with the following parameters: -MaxDist 500-MAF 0.05-Het 0.88-Miss 0.999.

### 2.5. Genome-Wide Association Analysis and Candidate Genes Identification

Associations between genotypic and phenotypic data were analyzed using the kinship matrix in an MLM (Mixed linear model) by GAPIT (http://www.zzlab.net/GAPIT/) based on R 3.6.1 to control background variation and eliminate the spurious MTAs. Due to Bonferroni-Holm correction for multiple testing (α = 0.05) was too conserved and only a few significant MTAs were detected with this criterion, markers with an adjusted –log_10_ (*p*-value) ≥ 6.0 were regarded as the significant ones.

Candidate genes for the loci consistently identified in two or more environments were identified. The following steps were conducted to identify candidate genes for important QTL. Firstly, found all the genes located in LD block region around the peak SNP (± 150 Kb based on LD decay analysis) of each important QTL from the MSU Rice Genome Annotation Project (http://rice.plantbiology.msu.edu/cgi-bin/gbrowse/rice/). Then, all available SNPs located inside of these genes were searched. The genes (except for expressed protein, hypothetical protein, transposon protein and retrotransposon protein) with SNP in coding region and which could further lead to sense mutation were considered as candidate genes. Besides, mesocotyl elongation is highly regulated by various phytohormones, including strigolactones (SLs), cytokinin (CTK), brassinosteroid (BR), abscisic acid (ABA), jasmonates (JAs), gibberellins (GA) and auxin (IAA). Thus, these genes involved in the metabolism of the phytohormones talked above were selected as high-confidence candidate genes for mesocotyl elongation.

### 2.6. Gene Expression Analysis

Quantitative real-time PCR (qRT-PCR) was conducted to test the expression differences of candidate genes between accessions with extreme mesocotyl length (Appendix A). The mesocotyl part was sampled at the 52 h after germination before the coleoptile excavated. Total RNA was extracted according to the Trizol method, cDNA was synthesized with the HiScript II 1st Strand cDNA Synthesis Kit (Vazyme, China). Primers (Appendix A) were designed with the primer premier 5.0 software (http://www.premierbiosoft.com/; last accessed Jan 2019). The PCR procedure was conducted in a volume of 20 μL, containing 2μL cDNA, 0.4 μL of each primer (μM), 10 μL ChamQ Universal SYBR qPCR Master Mix. *OsActin1* was used as the internal control to normalize the expression level of different samples. All assays were performed in two independent experiments, each with three repetitions.

## 3. Results

### 3.1. Phenotypic Variation of Mesocotyl Elongation

Mesocotyl length of 208 rice accessions were evaluated in water, sand and soil (2, 4 and 6 cm). Continuous variation was observed across all environments, presenting a wide range of mesocotyl length and indicating that this diversity panel was ideal for conducting GWAS (Figure 1). In sand culture, the mesocotyl lengths ranged from 0 to 5.03 cm, with an average of 1.67 cm; while in water, the mesocotyl lengths ranged from 0 to 3.57 cm, with an average of 1.21 cm. The seeds under soil were sown at a depth of 2, 4 and 6 cm, with the average mesocotyl lengths of 0.33 cm (0–0.71 cm), 0.51 cm (0–1.66 cm) and 2.47 cm (0–4.51 cm), respectively. Among different sowing depth, most mesocotyl lengths of 6 cm sowing depth were significantly longer than those of 2 cm or 4 cm sowing depth (Figure 1, Figure 2a). The maximum of mesocotyl lengths were 0.70 cm, 1.66 cm and 4.51 cm at 2 cm, 4 cm and 6 cm sowing depths in soil, respectively, indicating that mesocotyl elongation of these accessions had larger variation at 6 cm sowing depth than those at 2 cm or 4 cm. Furthermore, there was only a small proportion, about 32.4%, 7.2% and 70.0% accessions with longer mesocotyls (≥ 2.0 cm) in sand, water and soil (6 cm), respectively.

The correlation analysis showed that moderate correlations were observed among different culture media and high correlations were found among different sowing depth under soil. Mesocotyl length evaluated in sand showed active correlation with that under water (*r* = 0.70), soil (2 cm) (*r* = 0.74), soil (4 cm) (*r* = 0.75) and soil (6 cm) (*r* = 0.78). Also, mesocotyl length evaluated in water showed active correlation with that under soil (2 cm) (*r* = 0.76), soil (4 cm) (*r* = 0.78) and soil (6 cm) (*r* = 0.75). High correlations were observed among different sowing depth under soil. Mesocotyl length evaluated in soil (2 cm) showed significant active correlation with mesocotyl length at soil (4 cm) (*r* = 0.93) and soil (6 cm) (*r* = 0.85), whereas the correlation coefficient (*r*) between soil (4 cm) and soil (6 cm) was 0.86.

Analysis of variance (ANOVA) for mesocotyl length in 208 rice accessions revealed significant differences (*P* ≤ 0.001) among genotypes (G), environments (E), and genotype × environment interactions (GEI) (Table 1). The V_E_ is much larger than V_G_ and V_GEI_, indicating that mesocotyl length were greatly affected by environments. Besides, the broad sense heritability (*h*_b_^2^) estimate for mesocotyl length across all environments was 0.49. Thus, identifying the loci significantly associated with mesocotyl length under multiple environments is crucial for MAS breeding.

### 3.2. Marker Coverage

The sequencing data were mapped to the Nipponbare reference genome (IRGSP 1.0). After removing the SNPs with minor allele frequency (MAF) < 5% and missing data > 10%, 2,338,386 SNPs were left and employed for GWAS. The chromosome size varied from 22.8 Mb for chromosome 9 to 43.2 Mb for chromosome 1. These markers spanned a physical distance of 373 Mb, with an average density of 0.16 Kb per marker.

### 3.3. Population Structure and Linkage Disequilibrium

The diversity panel could be divided into two subgroups, the *Japonica* subpopulation and the other derived from the *Indica* accessions, whose characterization were largely consistent with geographic origins. Besides, admixture between *Indica* and *Japonica* were also observed in the present study (Figure 3c). Numerous studies have shown that the lack of appropriate correction for population structure can lead to spurious maker-trait associations (MTAs). Consequently, to eliminate spurious MTAs resulting from population structure, an MLM implemented in GAPIT were adopted for association analysis in the current study. PCA analysis indicated that the top three PCs could explain 28.5%, 8.2% and 3.2% of the total variation of population structure, respectively, and this panel consists of two subgroups (Figure 3a). The neighbor-joining (NJ) tree showing two clades also represented the two subpopulations (Figure 3b).

The decay of LD along physical distances was computed for the 208 rice accessions. A scatter *r*^2^ against physical distance showed a clean pattern of LD decay in the 208 rice accessions. A critical value of the determination coefficients *r*^2^ > 0.2 was determined to be the threshold for LD decay and the LD decay distance was about 150 Kb (Figure 3d).

### 3.4. GWAS of Mesocotyl Lengths

As Bonferroni correction was extremely conservative, a compromised threshold of –log_10_(*P*) ≥ 6.0 was used as the threshold for significantly associated SNPs. The Manhattan plots for the markers significantly associated with mesocotyl length under different conditions were shown in Figure 4. A total of 16 unique loci were detected, which explained the phenotypic variations ranged from 6.3% to 15.9%. Totally, 12 loci for mesocotyl length were identified on chromosomes 1, 2, 3, 5, 6, 8, 9 and 12 in sand, and each explained phenotypic variation ranged from 6.6 to 15.7%. Besides, seven loci were detected on chromosomes 2, 3, 5, 6 and 9 under water culture, which explain phenotypic variations ranging from 7.8 to 15.5%. There were remarkable differences among three sowing depths under soil culture. As far as the 2 cm sowing depth, a total of seven notable loci were detected on chromosomes 1, 4, 5, 7, 9 and 12, each explaining 7.6 to 15.7% of the phenotypic variation. When coming to the sowing depth of 4 and 6 cm, only two (chromosomes 1 and 12) and three (chromosomes 2, 6 and 12) loci were detected respectively, with an explanation rate of 14.4–14.8% and 6.3–15.9%.

Among all the above QTLs, 12 on chromosomes 1, 2, 3, 5 and 9 were identified under two or more environments. The loci located on chromosomes 1 (14.1–17.3 Mb) and 12 (13.3–15.0 Mb) were identified in sand, soil (2 cm) and soil (6 cm); whereas the locus on chromosome 12 (4.4-4.5 Mb) was detected in sand, soil (4 cm) and soil (6 cm). Besides, five loci on chromosomes 2 (5.6–8.7 Mb), 3 (15.2–15.3 Mb), 5 (5.8–6.3 Mb), 6 (7.3–9.7 Mb) and 9 (1.3–2.8 Mb) were identified in water and sand; two loci identified on chromosomes 5 (10.6–12.0 Mb) and 9 (7.1–9.1 Mb) were detected in sand and soil (6 cm); the locus on chromosome 6 (15.3–16.6 Mb) was identified on sand and soil (4 cm); the locus on chromosomes 2 (11.7–15.4 Mb) was identified on water and soil (4 cm) (Table 2, Appendix A, Figure 4).

### 3.5. Candidate Genes Related with Mesocotyl Elongation

According to the results of GWAS, 107 candidate genes were found to be related with mesocotyl elongation (Appendix A). Also, considering that mesocotyl elongation is highly regulated by various phytohormones, such as SLs, CTK, BR, ABA, JAs, GA and IAA, five genes involved in the biological metabolism of the phytohormones, cell elongation and division were selected as the high-confidence candidate genes for mesocotyl elongation (Table 3). That is the zinc finger CCCH type family protein (*LOC_Os05g10670*) and transcription factor jumonji (*LOC_Os05g10770*) on chromosome 5 (5.6–6.1 Mb), a gene encoding cytokinin-O-glucosyl transferase 2 on chromosome 9 (1.2–1.5 Mb) and two genes (4.3–4.5 Mb and 14.1–14.5 Mb) encoding flavin monooxygenase and 9-cis-epoxycarotenoid dioxygenase 1, respectively.

The expression of five candidate genes were detected using qRT-PCR (Figure 5). The results showed that there was significant difference between accessions with extreme mesocotyl length, with the longer one exhibited higher expression abundance. In detail, five genes showed more than 4.4-fold (*LOC_Os05g10670*), 2.8-fold (*LOC_Os05g10770*), 3.0-fold (*LOC_Os09g03140*), 8.0-fold (*LOC_Os12g08780*) and 32.6-fold (*LOC_Os12g24800*) higher expression in long-mesocotyl accessions compared to short-mesocotyl accessions, respectively.

## 4. Discussion

The GWAS panel, including released cultivars, advanced lines and landraces from different ecological regions, has a high genetic diversity with a wide range of mesocotyl length. In this study, *Indica* accessions had 32.4% and 70.0% accessions with longer mesocotyls (≥2.0 cm) in sand and 6 cm soil culture, respectively, but only 30.0% and 45.0% accessions in the *Japonica*, indicating that mesocotyl length had a great potential for improvement. Although mesocotyl is a crucial developmental trait of rice and is imminently required to improve crop adaptability to modern cultivation mode, the molecular mechanisms of its elongation and domestication remains unclear. Here, a GWAS based on re-sequencing were employed to identify associated SNP markers and understand the genetic basis of mesocotyl length.

### 4.1. Mesocotyl Elongation Varied among Culture Conditions

The elongation of mesocotyl was influenced by various environmental factors, such as light, temperature, moisture [28,29]. Thus, rice mesocotyl elongation were varied among different culture media, such as soil [26], sand [17], water [30]. In our study, mesocotyl length ranged from 0 to 3.99 cm under water culture, which was close to 0-5.05 cm in a previous study [17]. While, the mesocotyl length under 6 cm sand culture ranged from 0 to 4.88 cm, which differed from the previous result ranging from 0 to 2.05 cm under 5 cm sand culture [17]. Moreover, more than half of the 208 accessions had mesocotyl length longer than 1 cm under water culture. However, Wu et al. (2015) reported that only 10.5% accessions had mesocotyl length longer than 1 cm under water culture [17]. It is probably that the diverse results were mainly caused by different genotypes. Previous studies suggested that under varied sowing depths of soil cover, mesocotyl length was promoted by the depth in soil culture [11,18,26]. Lee et al. (1999) also reported that mesocotyl length under 3-cm sowing depth was shorter than that under 5-cm sowing depth [31]. Similarly, our results showed that mesocotyl length had a large variation among sowing depth, with an order of 6 cm > 4 cm >2 cm in soil. These results indicated that the variation of mesocotyl length existed in varied culture media and sowing depths.

### 4.2. LD Pattern

LD decay affects the precision of GWAS and was influenced by population structure, allele frequency, recombination rate and selection. Previous studies reported that LD decay in rice ranged between 120 Kb and 270 Kb [18,23,31,32]. In this panel, the LD decay was about 150 Kb for the whole genome (Figure 3e), consisting with previous reports [18,23,32,33].

### 4.3. GWAS Was Effective to Identify QTLs Controlled Complex Quantitative Traits

So far, several previous studies have reported the associated QTLs and candidate genes for rice mesocotyl elongation using linkage mapping or GWAS [20,25,34,35,36]. In this study, the loci identified on chromosome 7 (10.03–13.61 Mb) at the 6 cm sowing depth of soil was overlapped with QTLs *qFML7-1*, *qFML7-2*, *qFML7-3*, *qFML7-4* and *qFML7-5* [26]. The QTL on chromosome 1 (14.1–17.3 Mb) were overlapped with *qml1-1* [37,38] and *qML1* [23]. Furthermore, two QTLs on chromosome 2 (5.6–8.7 Mb) and 6 (7.3–9.7 Mb) identified under water and sand were overlapped with the QTL *qML2* and *qML6* detected from Shennong 265/Lijiangxintuanheigu RIL population [20]. Ouyang et al. (2015) have reported *qml3-1*, *qml4-1* and *qml5-1* by a RIL population [39], which nearly coincide with the QTL identified on chromosome 3 (15.2–15.3 Mb), 4 (8.3 Mb) and 5 (5.8–6.3 Mb) in our study. Zhang et al. (2006) detected four QTLs for submergence tolerance in seedling stage from a DH population, which explained phenotypic variations from 10.7% to 29.3% [35]. Among these, *qLOE-8* were overlapped with the QTL regions located on chromosome 8 (208–5.2 Mb) identified in sand. Lee et al. (2012) reported two QTLs associated with mesocotyl elongation, *qMel-1* (R^2^ = 37.3%) and *qMel-3* (R^2^ = 6.5%) in weedy rice on chromosomes 1 and 3, respectively [40]. Moreover, Lu et al. (2016) reported 17 loci and 289 candidate genes for rice mesocotyl by GWAS on chromosomes 1 (4), 4 (4), 5 (1), 6 (3), 7 (1), 9 (3) and 11 (1), respectively [18]. QTLs identified on chromosomes 9 (1.3–2.8 Mb) and 12 (13.3–15.0 Mb) were overlapped with those QTLs reported by previous studies [18,40].

Wu et al. (2015) found that when under water culture and at a significant level of –log_10_(P) ≥8.0, a total of 13 loci were detected to be significantly associated with mesocotyl length, while only three significant SNPs were declared when the medium was merely changed to sand, and two of the three SNPs were co-localized under both conditions [17]. Lu et al. (2016) reported that six and seven loci were detected under two environments, respectively [18]. Only two of them on chromosome 6 were identified under both environments. In different studies, nearly 40 loci significantly associated with mesocotyl length have been detected on all twelve chromosomes by GWAS, and most of these loci in each study cannot be validated by others [17,18,25,26]. The interaction of G (genotype) × E (environment) can influence QTL and association mapping results, indicating that the effect of identified genomic regions was required to be estimated for each environment [41,42]. Only the major genomic regions could be co-localized under multi-environments.

### 4.4. Potential Candidate Genes for Mesocotyl Elongation were Identified

According to the result of GWAS, 107 candidate genes were found to be related to mesocotyl elongation. Among them, five involved in the biological metabolism of phytohormones, cell elongation and division were selected as the high-confidence candidate genes. Bioinformatics analysis indicated that loci on chromosome 5 (5.6–6.1 Mb) corresponding to zinc finger CCCH type family protein (*LOC_Os05g10670*) and transcription factor jumonji (*LOC_Os05g10770*). CCCH-type zinc finger proteins comprise a large family that is induced by drought, high temperature stress and hydrogen peroxide, and is also induced by abscisic acid, methyl jasmonate and salicylic acid [43,44]. Noh et al., (2004) reported that jumonji class transcription factor controls stem elongation, transposon activity and panicle development [45]. Besides, Feng et al. (2017) have identified several candidate genes for mesocotyl in rice including zinc-finger protein genes, which involved in the JA biosynthesis and signaling pathways [29]. Besides, a candidate gene on chromosome 9 (1.2–1.5 Mb) encoding cytokinin-O-glucosyltransferase 2 were identified. CK is a class of plant hormones that were first identified as cell division-promoting factors and were subsequently identified as factors that control various processes in plant growth and development, including mesocotyl elongation [46]. Besides, CK played an important role in the biosynthesis of BRs, a group of steroid plant hormones essential in plant growth and development [47]. Two candidate genes in loci 4.3–4.5 Mb and 14.1–14.5 Mb on chromosomes 12 were identified, which encodes flavin monooxygenase and 9-cis-epoxycarotenoid dioxygenase 1, respectively. Flavin monooxygenase catalyzes hydroxylation of the amino group of tryptamines, a rate-limiting step in tryptophan-dependent auxin biosynthesis, which regulate many processes in plant development [48]. The 9-cis-epoxycarotenoid dioxygenase is essential for the biosynthesis of ABA [49]. The 9-cis-epoxycarotenoid dioxygenase gene could lead to the over-production of abscisic acid, which plays an important role in mesocotyl elongation [50]. Expression of the five candidate genes in different accessions indicated that five genes were all functional in regulating mesocotyl elongation. The most significant difference in expression level was observed in gene, *LOC_Os12g24800*, which indicated that this gene could be the most likely functional gene for mesocotyl length. These results also proved that the loci identified by GWAS were reliable. Mesocotyl elongation is a consequence of complicated biological processes and its mechanism remains unclear; more detailed experimental analyses are needed to confirm the function of candidate genes in mesocotyl elongation.

## 5. Conclusions

In the present study, a GWAS for mesocotyl length in 208 rice accessions was conducted. Mesocotyl length varied in different culture conditions and sowing depths. GWAS have identified 16 loci significantly associated with mesocotyl elongation, which explained phenotypic variations ranged from 7.1% to 10.0%. Furthermore, five high-confidence candidate genes were identified on chromosomes 1, 3, 5, 9 and 12. The accessions with longer mesocotyl and the markers significantly associated with mesocotyl length can be used to promote the process of rice breeding. This study will facilitate our uanderstanding on the genetic architecture of rice mesocotyl length.

## Figures and Tables

**Figure 1 genes-11-00049-f001:**
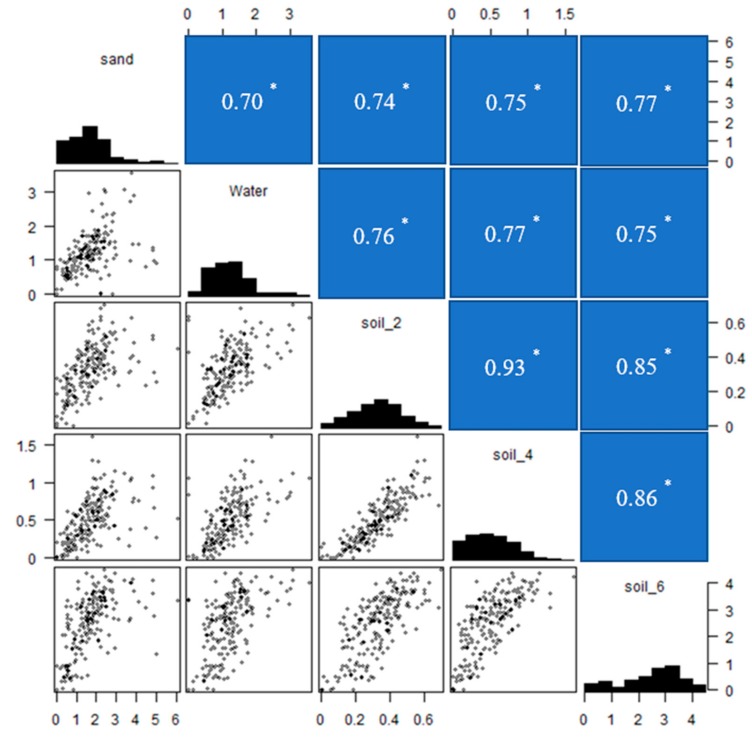
Summary of the phenotypic variations of mesocotyl length among different culture conditions, including correlations between mesocotyl lengths under different culture conditions and mesocotyl length distribution under varied culture conditions. Soil_2: soil culture with 2 cm sowing depth; Soil_4: soil culture with 4 cm sowing depth; Soil_6: soil culture with 6 cm sowing depth, numbers in blue boxes are correlation coefficients (*r*), * indicates the significance at *p* ≤ 0.05.

**Figure 2 genes-11-00049-f002:**
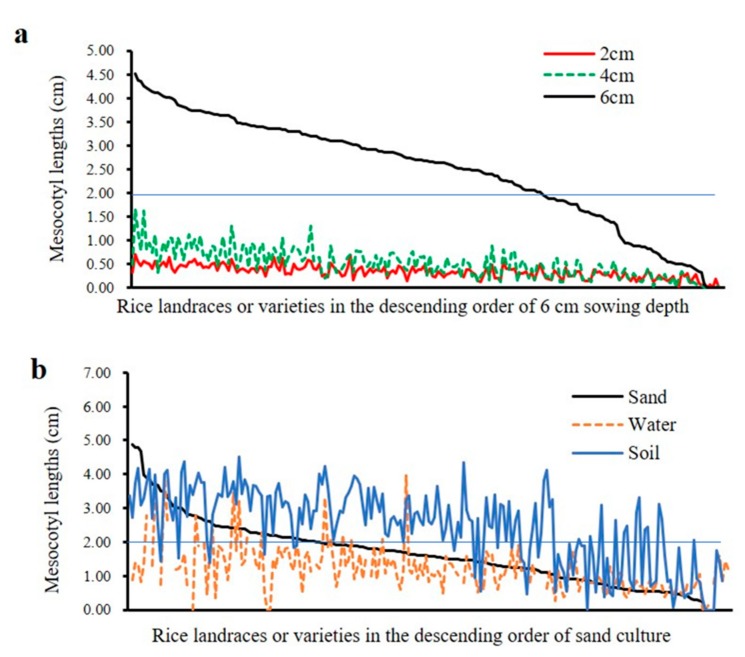
Phenotypic variations of mesocotyl elongation among three culture media. (**a**) Mesocotyl length of 208 rice accessions measured in soil culture with three sowing depths. (**b**) Mesocotyl length variation in sand, soil culture with 6 cm sowing depth and water.

**Figure 3 genes-11-00049-f003:**
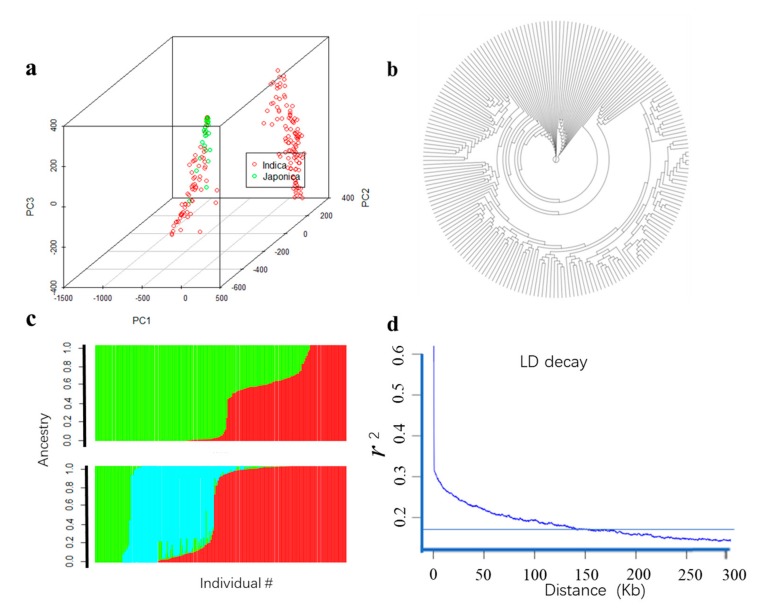
Population structure and LD decay of 208 rice accessions. (**a**) Principal component plot, different colors represent different ingredients; (**b**) neighbor-Jointing tree; (**c**) structure; (**d**) LD decay analysis.

**Figure 4 genes-11-00049-f004:**
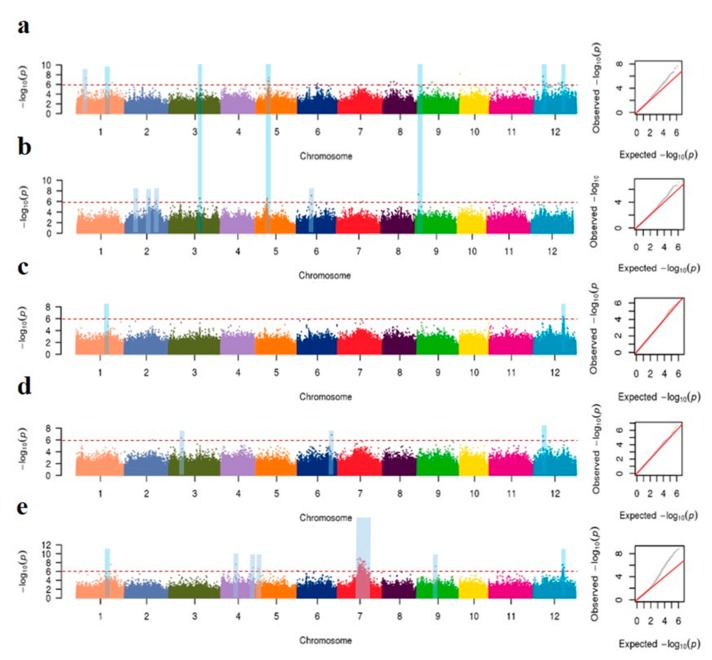
Genome-wide association studies for mesocotyl length among three culture media. Manhattan plots and Quantile-quantile plots of mesocotyl lengths in sand culture (**a**), water (**b**) and soil culture with a sowing depths of 2 cm (**c**), 4 cm (**d**) and 6 cm (**e**), respectively. Negative log_10_-transformed *p* values from a genome-wide scan are plotted against position on each of 12 chromosomes. The red horizontal dashed lines indicate the genome-wide significance threshold. Blue stripes show important signals in different GWAS.

**Figure 5 genes-11-00049-f005:**
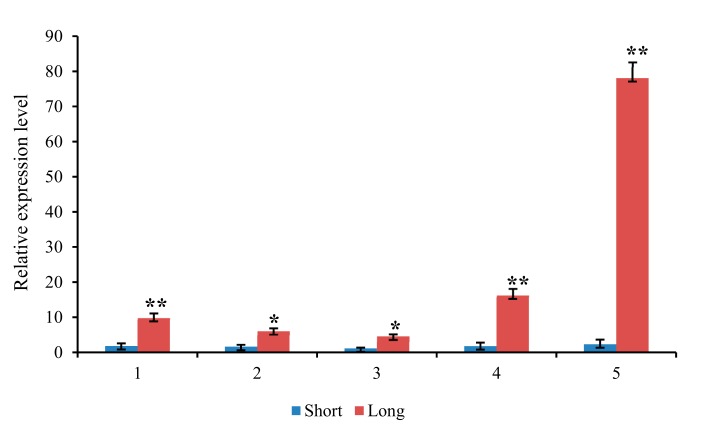
Expression analysis of five candidate genes. Relative expressions of the five genes in long- and short -mesocotyl length rice accessions were tested by qRT-PCR. Transcription levels were normalized to that of *Osactin1* and compared to the short control. 1 to 5 represent five candidate genes, that is 1, *LOC_Os05g10670*; 2, *LOC_Os05g10770*; 3, *LOC_Os09g03140*; 4, *LOC_Os12g08780*; 5, *LOC_Os12g24800*. Values of fold change are showed in mean ± SEM. Relative transcription levels were calculated by the log2^-^^△△CT^ method. * and ** indicate the significance at *p* ≤ 0.05 and *p* ≤ 0.01, respectively.

**Table 1 genes-11-00049-t001:** ANOVA for mesocotyl length under different environment.

Source	df	Sum of Square	Mean Square	*F*-Value
Block (Environment)	5	0.9063	0.1813	2.5698 ***
Genotype	207	583.1571	2.8172	39.9407 ***
Environment	4	1270.5662	317.6415	4503.3657 ***
Genotyped × Environment	813	533.9877	0.6568	9.3120 ***

*** indicates the significance at *p* ≤ 0.001.

**Table 2 genes-11-00049-t002:** Loci significantly associated with mesocotyl length under different environment.

Peak Marker	Chr.	Genomic Regions (Mb)	*P* Value	R^2^ (%)	Environment	Reported Locus	References
*rs_1_16752644*	1	14.1~17.3	1.9 E-08~7.7 E-07	7.2–15.2	Sand, Soil (2 cm), Soil (6 cm)	*qml1-1 and qML1*	Cao et al. 2002a, Huang et al., 2010
*rs_2_5647116*	2	5.6-8.7	4.8~9.4 E-07	7.6–8.1	Water, Sand	*qML2*	Huang et al., 2010
*rs_2_11740641*	2	11.7-15.4	2.8-7.3 E-07	8.1–15.5	Water, Soil (4 cm)		
*rs_3_15245138*	3	15.2-15.3	2.5-5.7 E-07	14.8–14.9	Water, Sand	*qml3-1*	Ouyang et al., 2005
*rs_4_7234284*	4	7.2	8.2 E-08	14.9	Soil (6 cm)	*Qml4-1*	Ouyang et al., 2005
*rs_5_6206136*	5	5.8-6.3	9.5E-09~9.7E-07	6.6–15.3	Water, Sand	*Qml5-5*	Ouyang et al. 2005
*rs_5_11950679*	5	10.6-12.0	6.6~7.6E-07	6.6–8.1	Sand, Soil (6 cm)		
*rs_6_7313005*	6	7.3-9.7	7.7E-08~8.4E-07	7.1–8.1	Water, Sand	*qML6*	Huang et al., 2010
*rs_6_16643928*	6	15.3-16.6	1.2~3.7 E-07	6.7–14.4	Sand, Soil (4 cm)		
*rs_7_10769188*	7	10.0-13.6	2.8E-09~9.9E-07	6.4–15.9	Soil (6 cm)	*qFML7-1*	Zhao et al., 2018
*rs_8_5219449*	8	2.8-5.2	6.8E-08~3.5E-07	14.6–15.3	Sand	*qLOE-8*	Zhang et al., 2006
*rs_9_1446824*	9	1.3-2.8	4.8E-08~4.8E-07	14.9–15.2	Water, Sand	*seq-rs4080*	Lu et al., 2016
*rs_9_7059246*	9	7.1-9.1	8.5 E-08~1.2 E-07	14.7–14.8	Sand, Soil (6 cm)		
*rs_12_4444693*	12	4.4-4.5	4.3E-09~5.9E-07	14.8–15.7	Sand, Soil (4 cm), Soil (6 cm)		
*rs_12_6008455*	12	6.0-7.8	1.2~4.6 E-07	14.7–15.4	Sand		
*rs_12_14155486*	12	13.3-15.0	2.3E-08~9.9E-07	7.5–15.3	Sand, Soil (2 cm), Soil (6 cm)	*qMel-12*	Lee et al. 2012b

**Table 3 genes-11-00049-t003:** High confidence candidate genes for mesocotyl lengths in 208 rice landraces or varieties.

Chromosome	Genomic Regions	Candidate Gene	Functional Annotation
5	5688534-6067598	*LOC_Os05g10670*	Zinc finger CCCH type family protein, putative, expressed
5	5688534-6067598	*LOC_Os05g10770*	Transcription factor jumonji, putative, expressed
9	1245230-1521824	*LOC_Os09g03140*	Cytokinin-O-glucosyltransferase 2, putative, expressed
12	4369693-4551492	*LOC_Os12g08780*	Flavin monooxygenase, putative, expressed
12	14080486-14520450	*LOC_Os12g24800*	9-cis-epoxycarotenoid dioxygenase 1, chloroplast precursor, putative, expressed

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
