# Peer review of "Genome-Wide Association Study (GWAS) for Mesocotyl Elongation in Rice (Oryza sativa L.) under Multiple Culture Conditions"

_genes, 2019, doi:10.3390/genes11010049_

Round 1

Reviewer 1 Report

In this article, the authors performed GWAS for mesocotyl length under various culture conditions. Several loci associated with mesocotyl elongation containing potentially novel loci were identified, and five candidate genes were shown expression difference between short- and long- mesocotyl accessions. These results are informative for the research community.

However, the present article has any problems to need improvement.

Major comments

1) In discussion, a part is considered incorrect.

In P13 L334, it is stated that “none of the SNPs was co-localized under both conditions.”, but by Wu et al. states that “two SNPs locating in the same regions associated to MLw on chromosome 3 and 6, one SNPs on chromosome 10 with no association to MLw.”

2) In P13 L296, it is described that “The elongation of mesocotyl was influenced... such a light, temperature, moisture.”. It could be worth checking various environmental factors in each culture conditions.

3) Related to the above, how do you consider the combination of environments when were identified loci for mesocotyl length under two or more environments. Is it related to the correlations were observed among different culture conditions?

I think it is important for this article.

4) In result, it is stated that “mesocotyl elongation had larger variation at 6 cm sowing depth than those at 2 cm or 4 cm.” It is insufficient discussion in this point. How about in addition to reference, e.g. Lee et al. 2002.

Minor comments

1) Figure 1, it would be not enough the figure legend, e.g. numbers in figure and the statistical processing.

2) In P6 L184, why are you on this condition (≥ 2.0 cm)? If it’s important, I would prefer to add a line to figure 2.

3) In figure 3, the fonts are too small to be legible.

4) In P7, there are not spelled out abbreviations, namely ‘MTAs’ and ‘NJ’.

5) In P7, the sub-numberings of figure 3 are better to be showed in sentences.

6) In Figure 4, it is better to indicate not only important signals, but also other detected signals.

7) In the legend of Figure 5, asterisks not explained.

I hope these comments will be helpful.

Reviewer 2 Report

This paper deals with GWAS for mesocotyl elongation in rice under multiple culture conditions, which has large impacts on the understanding of the genetic architecture of rice mesocotyl. The accessions with longer mesocotyl and the markers associated with mesocotyl length may certainly be of use for the process of rice breeding.

This paper can be accepted only after the revision of this paper, considering the comments below.

The letter r in Pearson r must be italicized. Please add information about the availability of the raw Illumina sequencing data from this study. Figure 1. Figure 5. Table 1. What meaning did the asterisk (*) refer to?
